# Assessment of residential exposures to agricultural pesticides: A scoping review

Raphaëlle Teysseire[1,2,3,4]*, Guyguy Manangama[1,2,3], Isabelle Baldi[1,2], Camille Carles[1,2], Patrick Brochard[1,2,3], Carole Bedos[5], Fleur Delva[1,2,3]

**1** Bordeaux Population Health Research Center, Inserm UMR1219-EPICENE, University of Bordeaux, Bordeaux, France, **2** Department of Occupational and Environmental Medicine, Bordeaux Hospital, Bordeaux, France, **3** Environmental health platform dedicated to reproduction, ARTEMIS center, Bordeaux, France, **4** Regional Health Agency of Nouvelle-Aquitaine, Bordeaux, France, **5** ECOSYS, INRA-AgroParisTech-Université Paris-Saclay, Thiverval-Grignon, France

* raphaelle.teysseire@chu-bordeaux.fr

**Data Availability Statement:** All relevant data are within the manuscript and its Supporting Information files.

## Abstract

The assessment of residential exposure to agricultural pesticides is a major issue for public health, regulatory and management purposes. In recent years, research into this field has developed considerably. The purpose of this scoping review is to provide an overview of scientific literature characterizing residential exposure to agricultural pesticides and to identify potential gaps in this research area. This work was conducted according to the JBI and PRISMA guidelines. Three databases were consulted. At least two experts selected the eligible studies. Our scoping review enabled us to identify 151 articles published between 1988 and 2019 dealing with the assessment of residential exposure to agricultural pesticides. Of these, 98 (64.9%) were epidemiological studies investigating possible links between pesticide exposure and the onset of adverse health effects, principally cancers and reproductive outcomes. They predominantly used Geographic Information Systems and sometimes surveys or interviews to calculate surrogate exposure metrics, the most common being the amounts of pesticides applied or the surface area of crops around the dwelling. Twenty-six (17.2%) were observational measurement studies conducted to quantify levels of pesticide exposure and identify their possible determinants. These studies assessed exposure by measuring pesticides in biological and environmental matrices, mostly in urines and house dust. Finally, we found only eight publications (5.3%) that quantified the risk to human health due to residential exposure for management purposes, in which exposure was mainly determined using probabilistic models. Pesticide exposure appears to be largely correlated with the spatial organization of agriculture activities in a territory. The determinants and routes of exposure remain to be explored to improve the conduct of epidemiological and risk assessment studies and to help prevent future exposures. Improvement could be expected from small-scale studies combining different methods of exposure assessment.

## Introduction

Pesticides are used to protect crops against undesirable organisms or diseases and also to influence the life processes of plants and conserve plant products [1]. They have been widely used

**Funding:** The authors received no specific funding for this work.

**Competing interests:** The authors have declared that no competing interests exist.

in agriculture throughout the world to manage natural hazards and ensure a high-yield food production for the past decades. Depending on the application method, compound mobility and persistence in the environment, pesticides can contaminate all environmental compartments, such as soils, water or air. Pesticides are generally applied in the fields by spraying and enter the atmosphere by different pathways. Direct emissions into the air can occur during application from spray drift, and indirect emissions happen post-application due to the volatilization of pesticides from plants and soils or wind erosion for several days or weeks after field application. Pesticides can eventually be transported across long distances, then form deposits on soils or surface waters, and then finally be transferred into ground waters [2]. These phenomena result in a general contamination of the different environmental matrices and the exposure of off-targeted environments and species, including humans.

Pesticide exposures and their health effects have mainly been studied among farmworkers. Associations between occupational exposure and several diseases such as cancers, neurological pathologies, and adverse effects on fertility or pregnancy have been demonstrated in the scientific literature [3]. Less is known about non-occupational exposures linked to agricultural treatments, although this has become an emerging research area in recent years. Due to their proximity to the fields, people residing in rural areas are the subject of greatest concern. Currently, there is no harmonized definition of the term "resident" in the scientific literature. In Europe, the European Food Safety Authority (EFSA) has defined residents as the persons who live, work or attend school near crop fields treated with pesticide and whose presence is unrelated to work involving pesticides, but whose position might lead them to be exposed [4]. Indeed, residents could potentially be exposed to pesticides due to spray drift during application and volatilization of pesticides after their application by pathways such as inhalation, dermal exposure or ingestion (via food or hand- or object-to-mouth transfer for toddlers) [4].

A good understanding of residents' pathways and levels of exposure to agricultural pesticides is essential for several reasons. First, an accurate measurement of the exposure is required in epidemiological studies in order to investigate possible links between the agricultural use of pesticides in the close environment and the onset of adverse health outcomes. Second, some pesticide regulations require a risk assessment on human health prior to the marketing of any plant protection products. In this evaluation process, acute and chronic exposures to pesticides are estimated quantitatively using several databases and models for comparison with toxicological values below which no risk to human health is expected. The USA regulation requires risk assessments to be carried out for the general population, taking into account all exposure pathways, including residential non-dietary exposures [5]. The European regulation requires a risk assessment for four exposed groups, including residents. Substances will be authorized on the market only if they are not expected to have any harmful effect on human [1]. Finally, exposure assessment is needed in order to propose preventive operational measures to protect populations, including routine monitoring of pesticides in the air.

Given these major issues, it seemed interesting to give an overview of why and how pesticide exposure of residents has been studied so far in the scientific literature. A partial inventory of approaches assessing residential exposures has previously been conducted in two articles, published in 2005 and 2015 [6,7]. The results showed that exposures to agricultural pesticides had mainly been approximated by spatialized indicators generated using Geographic Information Systems (GIS). Exposure has been measured to a lesser extent by biometrology, mostly in urines, and environmental monitoring in air and house dust. To provide a more complete overview of the existing scientific literature on the subject it is essential to update this data, as this field of research is constantly expanding, and to broaden the proposed framework by including not only epidemiological studies but also risk assessment studies or methodological developments. It also seemed interesting to collect information about the temporal and

geographical distribution of the publications. Given the purpose of this article, a scoping review was determined to be an appropriate research method. The three specific objectives of our study were (1) to provide an overview of studies assessing residential exposure to agricultural pesticides, (2) to describe the measurement methods used to characterize these exposures and (3) to identify potential gaps in this research area.

## Method

### General framework for review

We first established a research group with experts in epidemiology and expology in the field of pesticides. The scoping review was conducted using the methodological steps outlined in Arksey and O'Malley's (2005) framework combined with the enhancements by Levac et al. (2010) and Colquhoun et al. (2014) [8–10]. We followed the first five steps of this six-stage framework: 1) identifying the research question, 2) identifying relevant studies, 3) study selection, 4) charting the data, 5) collating, summarizing and reporting the results. The sixth level, which consists of a consultation with stakeholders, is optional and was not included in this study. To provide clarity and transparency in our approach, our work is presented following the guidelines proposed by the Joanna Briggs Institute (JBI) and the Preferred Reporting Items for Systematic Reviews and Meta-Analyses Extension for Scoping Reviews (PRISMA-ScR) [11,12].

### Research question

Our review was driven by the following question 'Why and how have residential exposures to agricultural pesticides been assessed in scientific studies?'

### Search strategy

The search for articles was carried out using three different online bibliographic databases: PubMed, Web of sciences and Scopus. We used the following algorithm: (pesticide* OR fungicide* OR herbicide* OR insecticide*) exposure* AND ((proximity AND (fields OR crop* OR agricultur*)) OR (residen*) OR ((agricultural OR rural) AND (communit* OR area*))). We also included articles identified from websites or in the bibliographic references of the selected studies. The references were managed using Mendeley Reference Manager [13].

### Study eligibility criteria

Our a priori inclusion criteria were:

1. human studies published in English, French or Spanish.

2. studies explicitly mentioning pesticides used in agriculture.

3. studies including a 'spatialized' definition of the term 'resident'. Thus, only studies that provided numerical spatial indicators such as the residential distance from the field or the agricultural use of pesticides in the vicinity were included. A qualitative definition such as "rural area" was not considered as sufficiently precise to define residents. We did not consider a maximum distance to define residents.

4. studies including an assessment of residential exposures.

As a scoping review is an iterative method, we chose to modify our eligibility criteria *post hoc* by excluding studies where pesticides were applied by plane or helicopter. The levels and profiles of exposure are certainly very different in the populations concerned. Studies including only farmers' families were also excluded. There is already sufficient evidence to show that

farmers' children and spouses are more exposed to pesticides than the general population, due to the take-home exposure pathway [14,15] and it could be difficult to study their exposures following outdoor agricultural treatments. However, studies including famers' families and also other people living near the crops without agricultural activities have been retained. Finally, we excluded epidemiological publications focused on clusters of cases.

Three reviewers participated in the selection of the relevant studies (RT, GM, FD). The eligibility of each article was determined by two reviewers independently. In the event of disagreement, a consensus was found between all the reviewers about the status of the article.

## Data collection and analyses

We built a database to enter the relevant information in a standardized survey including the item of interest. The form included six sections: 1) global information about the article (reference, year of publication, geographic area), 2) study population (type and size of population, pathologies studied), 3) type of crops and pesticides of interest, 4) pollutants' transfer routes (drift, volatilization) and exposure pathways considered (inhalation, ingestion, dermal exposure), 5) method used to assess the exposure (metrology, modelling, surveys, etc.), 6) Key findings. The data were entered and analyzed using Access and Excel 2010.

## Results

### Literature search

The search for articles in the three databases was carried out on October 10, 2018 and was continued until October 2019 to identify other sources. The different steps of the study selection process are detailed in the flow diagram in Fig 1.

Finally, 151 articles matched our inclusion criteria. The complete list of the selected publications is available in S1 Appendix. The principal reasons for exclusion were: no mention of pesticides used in agriculture, absence of human population and inclusion of populations other than residents (workers or general population).

### General characteristics of included studies

The general characteristics of the studies analyzed are presented in Table 1. These articles were published between 1998 and 2019. There was a continuous increase in the number of publications per year, with a marked rise since the 2010's. North America and Europe produced the majority of the articles published (90.7%).

In our analysis, we distinguished three types of study according to their main objectives: epidemiological, observational measurements and risk assessment. We defined "observational measurement studies" as non-experimental studies that characterized residents' exposure to pesticides using metrology but which did not investigate health effects related to these exposures.

Epidemiological studies were predominant (n = 98, 64.9%) compared with observational measurement studies (n = 26, 17.1%) and risk assessment studies (n = 7, 5.3%). Nineteen studies (12.6%) developed frameworks to assess residents' exposure in epidemiological and risk assessment studies. Key data devoted to exposure assessment collected in each type of study are described in the following parts.

### Assessment of residential exposures in epidemiological studies

Epidemiological studies represented the majority of articles that assessed residential exposure to agricultural pesticides. Their main objective was to investigate the link with one or several

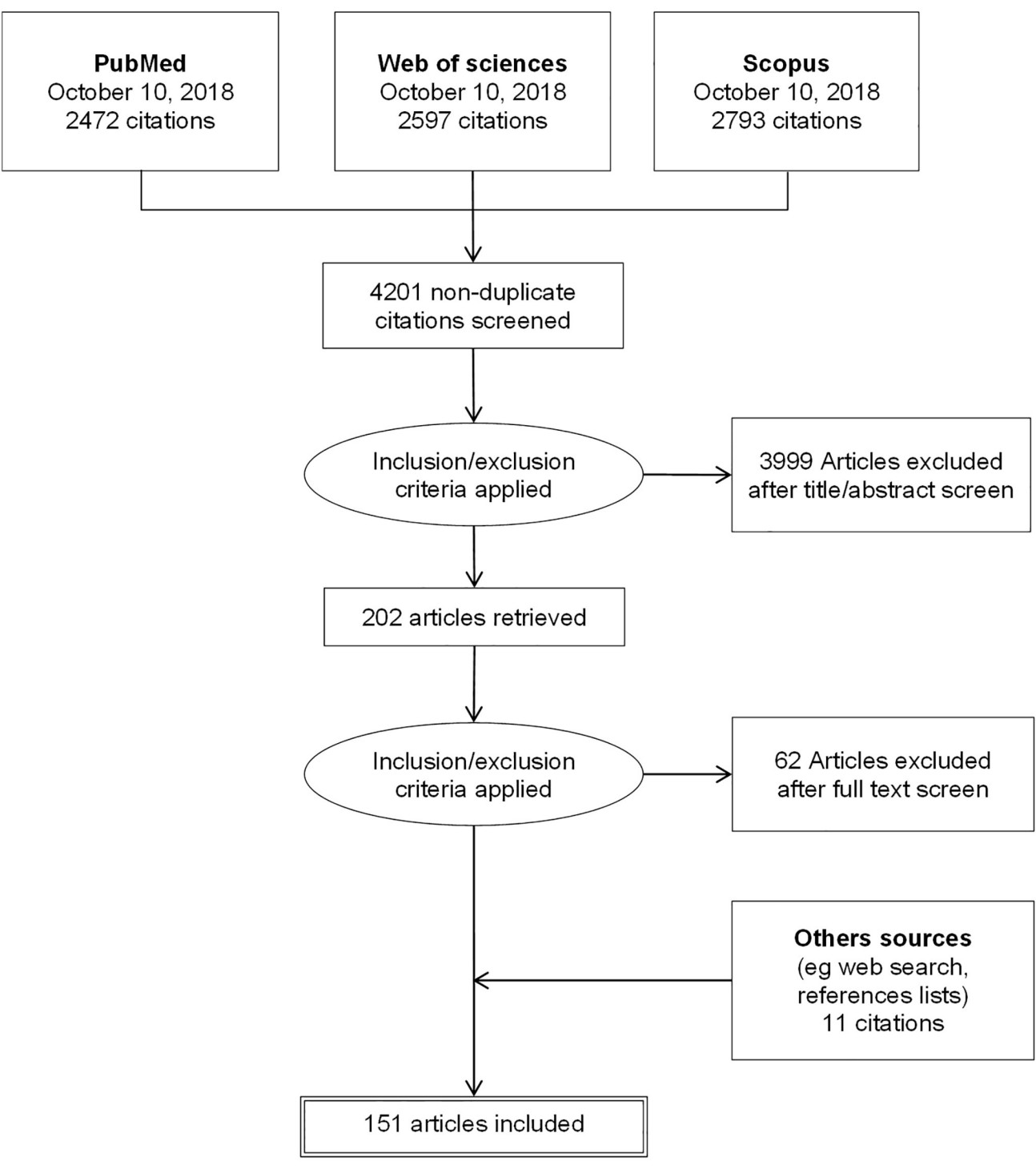

**Fig 1. Flow diagram of study selection.**

health outcomes. The main health outcomes under study were cancers (n = 33, 33.7%), congenital malformations (n = 21, 21.4%), neurological diseases (n = 14, 14.3%), child development (n = 13, 13.3%), pregnancy complications (n = 10, 10.2%), genetic or epigenetic modifications (n = 5, 5.1%), respiratory diseases (n = 3, 3.1%) and others such as overall

**Table 1. General characteristics of studies about residents' exposure to agricultural pesticides published between 1998 and 2018 included for scoping review (n = 151).**

|  | n (N = 151) | % |
|---|---|---|
| Study type |  |  |
| Epidemiological studies | 98 | 64.9 |
| Case-control study | 56 | 37.1 |
| Ecological study | 19 | 12.6 |
| Cohort study | 17 | 11.3 |
| Cross sectional study | 4 | 2.6 |
| Protocol for future cohort study | 2 | 1.3 |
| Methodological framework for epidemiological studies | 8 | 5.3 |
| Observational measurement studies | 26 | 17.2 |
| Risk assessment studies | 19 | 12.6 |
| Risk assessment studies | 8 | 5.3 |
| Methodological framework for risk assessment studies | 11 | 7.3 |
| Publication year |  |  |
| < 2000 | 4 | 2.6 |
| 2000–2004 | 21 | 13.9 |
| 2005–2009 | 25 | 16.6 |
| 2010–2014 | 37 | 24.5 |
| 2015–2019 | 64 | 42.4 |
| Geographic zone |  |  |
| North America | 91 | 60.3 |
| Europe | 46 | 30.4 |
| Asia | 5 | 3.3 |
| South America | 4 | 2.6 |
| Africa | 3 | 2.0 |
| Central America | 1 | 0.7 |
| Australia | 1 | 0.7 |

mortality, biological effect (blood pressure, acetylcholinesterase activity), and acute pesticide intoxications (n = 4, 4.1%). Details of all the health effects studied in the selected publications are available in S2 Appendix. Populations recruited in these epidemiological studies were predominantly pregnant women (fetus) or children (n = 62, 63.3%). Population sizes varied greatly depending on the study design.

A wide range of active substances were included, with a quasi-equal representation of herbicides (n = 35, 35.7%), insecticides (n = 32, 32.7%) and fungicides (n = 30, 30.6%). The number of substances included could be very different from one study to another, with a mean of 179 over a range of 1 to 850 pesticides. Most studies (n = 69, 70.4%) did not specify the kind of crops in the study area. When information was available, they were arable crops (n = 20, 20.4%), fruits (n = 12, 12.2%), vegetables (n = 11, 11.2%), flowers and bulbs (n = 8, 8.2%), vineyards (n = 7, 7.1%), or other crops (n = 2, 2.0%).

Exposure assessment methods used in these studies are presented in Table 2. Various approaches, possibly combined, were used to assess exposure. Direct measuring such as biological and/or environmental monitoring was rare (n = 4), while indirect methods were predominant, 99.0% (n = 97) of the epidemiological studies calculated spatialized exposure indicators using GIS or survey/interview with the subjects. In several studies, the principal indicator used was the amount of pesticides applied to the area of residence (n = 55; 56.1%). Numerous studies (n = 43, 43.9%) that calculated this parameter were conducted in California

**Table 2. Assessment of residential exposure in epidemiological studies (n = 98).**

| | n (N = 98) | % |
|---|---|---|
| Assessment of agricultural pesticide exposure | | |
| Biological monitoring | 4 | 4.1 |
| Urinary concentrations of pesticides | 2 | 2.0 |
| Concentrations of pesticides in the hair | 1 | 1.0 |
| Blood levels of acetylcholinesterase | 1 | 1.0 |
| Environmental monitoring | 1 | 1.0 |
| Pesticide concentrations in ambient air | 1 | 1.0 |
| Spatial surrogate for pesticide exposure | 97 | 99.0 |
| Amount of pesticides used in the area of residence | 55 | 56.1 |
| Surface area of crops in the area of residence | 25 | 25.5 |
| Complex score based on several parameters (distance, surface area, use of pesticides, meteorological data, etc.) | 8 | 8.2 |
| Residential distance from crops | 7 | 7.1 |
| Presence of crops in a defined perimeter | 1 | 1.0 |
| Financial agricultural productivity by county | 1 | 1.0 |
| Population size | | |
| med (min–max) | 1473 (48–25 110 289) | |
| Exposure period | | |
| Preconception period | 2 | 2.0 |
| Prenatal period | 46 | 46.9 |
| Childhood | 24 | 24.5 |
| Adulthood | 35 | 35.7 |
| Exposure duration (years)[1] | | |
| med (min–max) | 7.5 (< 1–50) | |
| Interval including the smallest distance or buffer radius around the residence considered in the spatial analysis[2] (meters) | | |
| [0–100] | 7 | 7.1 |
| [100–500] | 28 | 28.6 |
| [500–1000] | 19 | 19.4 |
| [1000–5000] | 7 | 7.1 |
| Number of substances studied[3] | | |
| med (min–max) | 22 (1–850) | |

[1] 3 unavailable data,

[2] 35 unavailable or irrelevant data,

[3] 55 unavailable or irrelevant data

using the Pesticide Use Report (PUR) data set. In California, all agricultural and non-agricultural pesticide uses are reported to the California Department of Pesticide Regulation. Information about crops, active substances, date and amount of pesticides applied, location of application, are compiled into the PUR data set with a resolution of 1.6 km (one mile) square sections [16].

Most of the studies that used spatial exposure surrogates considered only one distance or buffer radius around the house in their analyses (n = 57, 33.0%), while a minority (n = 19) included two to four perimeters in order to observe the influence of this metric on the occurrence of health outcomes. In GIS, a buffer is a zone specified around a point or a line or a polygon area. In all the studies selected, this zone was a circular polygon with a center determined

by the residential location. Distances or buffer radiuses considered ranged from 0 to 8,000 meters. The smallest metric around the residence considered in the analysis (resolution) was generally less than 500 m (n = 35, 57.4%).

Finally, 70 epidemiological studies (71.4%) reported a significant association between at least one of the health outcomes studied and residential exposure to pesticides. Of these, 7.1% concluded that there was a significant association for a distance from the residence to the source of less than or equal to 100 m, 24.3 for a distance less than or equal to 500 m, 14.3 for a distance less than or equal to 1,000 m and 4.3% for a distance greater than 1,000 m. This parameter was unavailable for the rest of these studies (n = 35, 50.0%).

Besides the epidemiological studies presented above, we found eight studies presenting methods to assess exposure in epidemiological studies. Two European articles and five American publications described methods to characterize environmental pesticide exposure using GIS. They principally studied the possibility of using GIS to calculate the amounts of pesticides applied in a perimeter around the residential location as a surrogate for non-dietary exposures (n = 5) [17–21]. One study chose to focus on the surface area of the agricultural land around the dwellings [22] and a second used the distance of the residence from the fields, but taking into account the possible presence of obstacles like forests [23]. The buffer radiuses around the residence varied considerably but they were all less than 1,000 meters. These studies concluded that using a GIS-model to assess residential exposures to agricultural pesticides was a feasible approach. We found one study that used an atmospheric dispersion model called CAREA to estimate residential exposures to agricultural pesticides for epidemiological purposes. CAREA is a GIS-based Gaussian model based on a simplification of AERMOD, a model developed by the US Environmental Protection Agency (EPA). The results from this model were compared to those obtained with a GIS-based proximity model. In this study, 2,584 people were considered as receptors. The authors concluded that the use of the atmospheric dispersion model led to a considerable increase in the percentage of exposed receptors, from 4% obtained with the proximity model to 54% with CAREA. A test on a specific site showed that the effects of meteorology considered by the atmospheric dispersion model led to an anisotropic exposure distribution around the emission source resulting in a slight underestimation/overestimation of the receptors' exposure.

## Assessment of residential exposures in observational measurement studies

We found 26 observational studies characterizing residential exposure to agricultural pesticides. Four small-scale observational studies measured pesticide exposure during actual applications. These studies were conducted in association with local farmers to define the nature of the substances applied and the spraying locations. Five large-scale studies conducted in California obtained this information from the PUR dataset. The rest of the observational studies did not report actual pesticide applications, but a majority measured pesticide exposure during a spraying season. Nine of the selected publications (34.6%) were based on measurements of pesticides in biological matrices, 11 (42.3%) on environmental measurements, while six (23.1%) combined the two approaches. Table 3 presents the measurement strategies developed in each study. Studies that carried out biological monitoring searched for metabolites of Organophosphorus compounds (n = 8) and Pyrethroids (n = 4), Triazine (n = 1), Carbamates (n = 1) and Azoles (n = 1) in pesticides. Two studies searched for manganese (Mn) levels in urines or deciduous teeth as an indirect measure of Mn-based fungicides such as Maneb or Mancozeb. Finally, one study investigated urinary metabolic profiles rather than pesticide residues. Studies carrying out environmental monitoring focused on one or several categories of pesticide such

**Table 3. Assessment of exposure in observational measurement studies.**

| | n (N = 26) | % |
|---|---|---|
| Biological monitoring | 15 | 57.7 |
| Biological matrices investigated | | |
| Urines | 13 | 50.0 |
| Hair | 1 | 3.8 |
| Blood/serum | 2 | 7.7 |
| Placenta | 1 | 3.8 |
| Deciduous teeth | 1 | 3.8 |
| Population | | |
| Children | 11 | 42.3 |
| Pregnant women | 4 | 15.4 |
| Adults | 3 | 11.5 |
| Population size | | |
| med (min–max) | 192 (20–1077) | |
| Number of substances included | | |
| med (min–max) | 5 (0–540) | |
| Number of measuring campaigns | | |
| med (min–max) | 1(1–11) | |
| Environmental monitoring | 15 | 57.7 |
| Environmental matrices investigated | | |
| House dust | 14 | 53.8 |
| Outdoor air | 3 | 11.5 |
| Indoor air | 2 | 7.7 |
| Hand wipe | 3 | 11.5 |
| Surfaces | 2 | 7.7 |
| Concentrations in grass samples | 1 | 3.8 |
| Population | | |
| Children | 9 | 34.6 |
| Pregnant women | 3 | 11.5 |
| Adults | 7 | 26.9 |
| Number of substances included | | |
| med (min–max) | 7 (2–46) | |
| Type of experimental sites included (n = 15) | | |
| Schools | 1 | 3.85 |
| Dwellings | 15 | 57.7 |
| Number of experimental sites included | | |
| med (min–max) | 96 (2–378) | |
| Number of measuring campaigns | | |
| med (min–max) | 1 (1–4) | |
| Interval including the smallest distance or perimeter around the residence considered in the spatial analysis[2] (meters) | | |
| [0–100] | 15 | 57.7 |
| [100–500] | 4 | 15.4 |
| [500–1000] | 1 | 3.8 |
| [1000–5000] | 3 | 11.5 |

[1] One study investigated urinary metabolic profiles rather than pesticide residues (number of substances included = 0),

[2] Three unavailable data

as insecticides (n = 11), fungicides (n = 8) and herbicides (n = 7). Two studies investigated manganese (Mn) levels in dust as a marker of fungicide contamination.

All the selected studies (n = 26) investigated the influence of indicators based on agricultural land-use on the global exposure to pesticides and 87.5% (n = 23) investigated the contribution of other parameters. Their results are shown in Fig 2. Finally, 76.9% (n = 20) of publications found a significant association between levels of pesticides in biological or environmental matrices and one or more spatial indicators, like the distance of residences or schools from the fields (n = 15), amounts of pesticides used in the vicinity (n = 5) and agricultural land surface areas in the vicinity (n = 1). The construction of the last two indicators requires the definition of buffers. More than half of the studies (n = 16, 66.7%) considered more than one distance or buffer radius around the residence location in order to observe the influence of source proximity on receptor exposure. Data on spatial resolution was unavailable for two articles. The smallest distance or buffer radius around a residence defining the zone of influence for agricultural pesticides was between 25 and 3,000 meters. Among the other parameters investigated, farmworkers living in the house, season and dietary consumption were frequently identified as contributing substantially to global pesticide exposure.

## Assessment of residential exposures in risk assessment studies

The purpose of risk assessment studies is to assess the probability that the health effects of a substance will occur in humans under specific exposure scenarios. Eight studies that conducted a risk assessment for residents, mostly around fruit crops (50.0%), were identified in our review. Four studies considered real residential and/or workplace locations. The source of exposure was generally chosen close to the receptors with four studies selecting a distance less than 50 m from the emitter.

One study used urinary concentrations of pesticides to estimate the internal dose of exposure, whereas the rest used measured environmental concentrations (n = 2) or exposure surrogates obtained using models (n = 4). Parameters included in the exposure assessment are presented in Table 4.

Five studies (62.5%) concluded there was a risk for human health linked to the proximity of agricultural pesticide applications, but only for a small proportion of pesticides, crops, and spraying conditions. Two of these studies identified dermal absorption as the main exposure route [24,25].

In addition to risk exposure assessment studies presented in Table 4, eleven articles presented models for risk exposure assessment. Most of them (n = 6) described the BROWSE model. This model has been developed by the European BROWSE project to predict the exposure of operators, workers, bystanders, and residents to agricultural use of pesticides [26]. The aim of this model is to improve regulatory exposure assessment by including recent data and changes in current knowledge and application practices. The BROWSE model has been developed in part using other preexisting models like PEARL, or OPS like those described by other authors [27]. The model uses a probabilistic approach to human health risk assessment based on the distribution of the estimated exposure doses from inhalation, dermal contact and ingestion pathways. The range of distances chosen to define residents is 2–20 m. This model is described as flexible and the model input can be widely fixed in order to simulate a panel of exposure scenarios. Although it was built for regulatory purposes, we found one risk assessment study that used this model to compare agricultural practices [24].

In addition to the BROWSE model, the EFSA guidance for determining non-dietary exposures of humans to plant protection products presents other probabilistic bystander and resident exposure assessment models built for regulatory purposes [4]. Some of them are

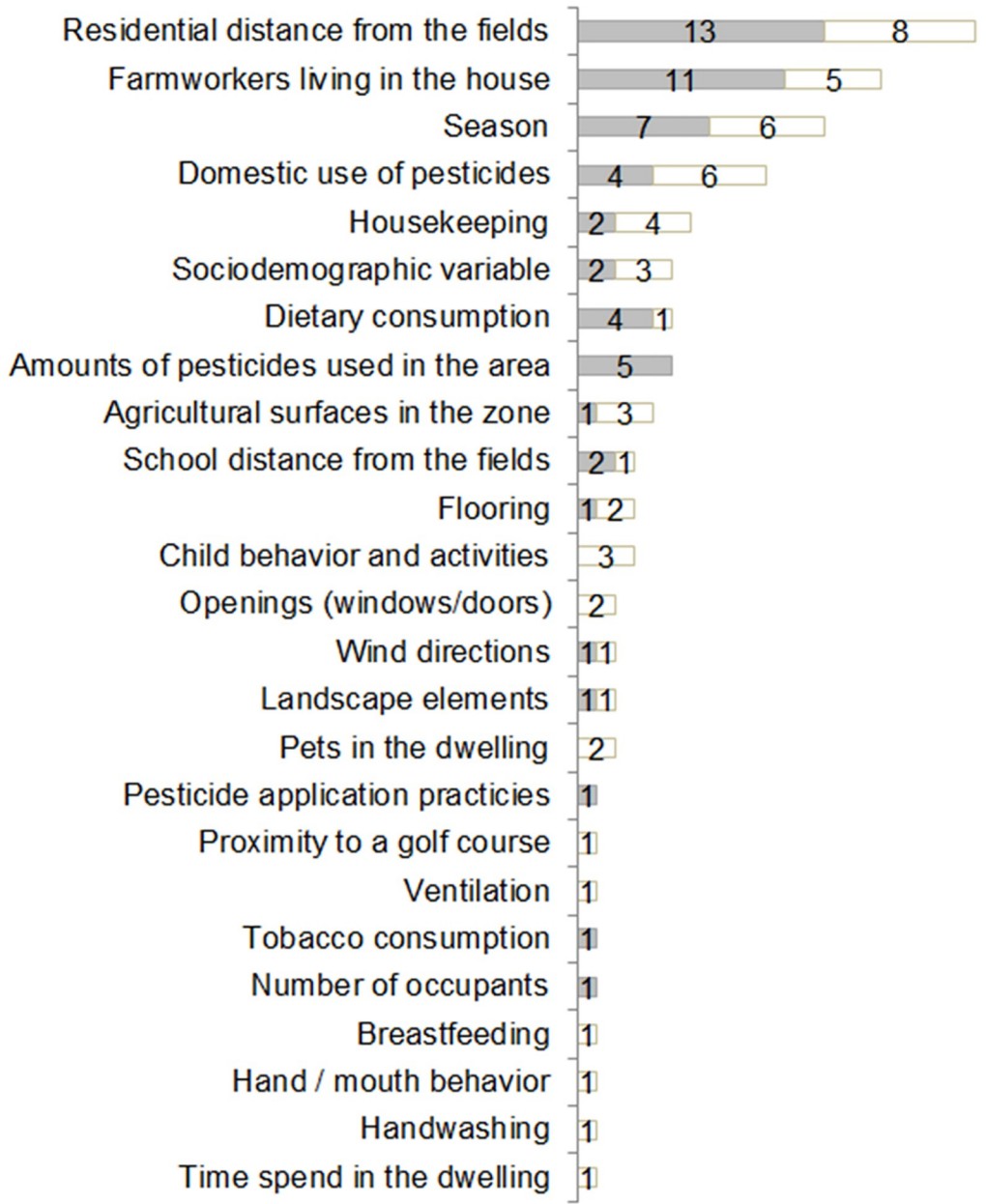

**Fig 2. Determinants of global exposure to pesticides explored in observational exposure studies (n = 26).**

discussed [28,29]. One article described a model not mentioned by EFSA estimating aggregate exposure (e.g. dietary and non-dietary sources) by combining some preexisting European models [30]. One study compared outputs from a regulatory model used in the UK with urinary biomarkers of five pesticides obtained from an experimental campaign. The study found

**Table 4. Exposure assessment in risk assessment studies.**

| | n (N = 8) | % |
|---|---|---|
| Assessment of environmental pesticide exposure[1] | | |
| Probabilistic or deterministic models | 3 | 37.5 |
| GIS-based tool | 1 | 12.5 |
| Environmental monitoring | 2 | 25.0 |
| Biological monitoring (urines) | 1 | 12.5 |
| Type of crops studied[1] | | |
| Fruit crops | 4 | 50.0 |
| Arable crops | 2 | 25.0 |
| All crops combined | 2 | 25.0 |
| Number of substances included | | |
| med (min–max) | 11.5 (1–132) | |
| Pesticide transfer pathway into the environment considered [2] | | |
| Drift only | 2 | 25.0 |
| Drift, volatilization and deposits | 4 | 50.0 |
| Exposure pathway considered | | |
| Dermal contact | 3 | 37.5 |
| Inhalation | 3 | 37.5 |
| Ingestion | 2 | 25.0 |

[1] 1 missing datum,

[2] 2 missing data

that almost all of the measures were lower than the level predicted by the model and concluded that the model was sufficiently conservative [31].

Finally, we found one article presenting a model built for anything other than regulatory use. This model focusing on spray drift was built to estimate inhalation exposure of residents in life-cycle assessment in the agrifood sector. The results showed that residential exposure was limited compared to dietary exposure due to the ingestion of pesticide residues in crops but that it could be substantially higher than the exposure of populations not living near the fields [32].

## Discussion

The main objective of our scoping review was to provide an overview of why and how residential exposure to agricultural pesticide has been studied so far in the scientific literature. Our study showed that this topic has become a growing area of research in the past few years with a significant increase in the number of publications since the 2010's. Most of the studies were epidemiological studies investigating possible links between pesticide exposure and the onset of one or several adverse health effects, principally cancers and reproductive outcomes. We found a smaller number of observational measurement studies conducted in order to quantify levels of pesticide exposure and identify its possible determinants. Finally, we found only a few publications that quantified the risk to human health due to residential exposure for management purposes. Two kinds of method were used to assess environmental exposure to pesticides: 1) direct approaches, including measurement in biological and environmental matrices that have been equally used in observational measurement studies, and 2) indirect methods, often based on modelling, that have been largely mobilized by epidemiological and risk

assessment studies to determine environmental concentrations or spatialized indicators as a surrogate for pesticide exposure.

## Evidence of agricultural drift pathway

According to observational measurement studies carried out in the past few years, residential exposure to pesticides seems largely determined by the spatial organization of agricultural activities on the territory. Thus, in the selected studies, the exposure of people residing near crops or living in intensive farming areas tended to be higher than that of people living with no agricultural activities nearby. Exposure also seemed to be influenced by the seasons, with higher levels of pesticide exposure recorded during agricultural sprayings.

Despite these observations, it is not possible to give a definition of the term "resident" based on a distance from the sources of agricultural emissions. According to the results obtained in the different studies selected in this analysis, the maximum distance defining the influence of agricultural emissions could be in the range of 25 to 3,000 meters. Most of the measurement studies explored only one perimeter around the residential location and possibly underestimated the real perimeter of the influence of pesticides. In addition, several parameters have been identified in the scientific literature as intervening in the spatial and temporal dispersion and transport of pesticides into the atmosphere, such as the physicochemical characteristics of the substances, the type of vegetation cover, the agricultural spraying equipment and the environmental conditions such as meteorological and topological parameters [2]. The significant variations in protocols from one study to another in terms of substances and crop selection, measurement strategy and geographic conditions could largely explain the wide range of distances observed. Variations exist between countries in spraying technologies, treated areas, types of pesticides authorized and used. These parameters may have also changed over time within a given country.

## Routes and determinants of residential pesticide exposure

The results of the observational measurement studies do not clearly identify the contribution of the different routes of exposure, nor its non-spatial determinants. House dust was the principal matrix explored in the observational exposure studies. Concentrations of pesticides in carpets were seen as a good proxy for residential long-term exposure because these chemicals are protected from degradation linked to external conditions [33]. However, it still remains an indirect measure of exposure, which does not differentiate the various non-dietary exposure pathways. Other matrices like indoor and outdoor airs were explored less, making it difficult to distinguish inhalation, ingestion, and dermal contact pathways. A few determinants other than spatial indicators were considered in exposure studies, such as subjects' characteristics, activities of the occupants or meteorological and topological conditions. Only the number of farmworkers living in the house was well identified as a major contributor to pesticide exposure, mainly because of the take-home pesticide exposure pathway, as already demonstrated in previous reviews [15,34]. Further small-scale measurement studies, at landscape level, could provide a better understanding of the pathways and determinants of pesticide exposure, which are important elements to refine exposure assessment in epidemiological studies or for proposing preventive measures for the population in environmental health policies.

## Assessment of residential exposure in epidemiological studies

For the most part, the epidemiological studies used indirect methods to evaluate pesticide exposure related to agricultural activities in the vicinity for groups of subjects in ecological studies and some case-control studies, or at an individual level for other study designs. A large majority used GIS to calculate spatial surrogates for pesticide exposure and a minority

employed surveys to estimate residential distance from the first fields. Only two studies used a dispersion model to assess concentrations in the vicinity [35,36]. The predominant use of GIS could be explained by the relative simplicity of these tools and the possibility of applying them to a data set including many observations [37]. Conversely, biological and environmental monitoring can be technically difficult or very expensive to deploy when the number of subjects increases. Some countries have interesting large-scale georeferenced datasets on the use of pesticides over long periods, such as the PUR data set implemented in California [16]. Developments are being deployed in other countries to acquire this type of data. In France, for example, data on the sale of pesticides to distributors are collected in a publicly accessible database. The forthcoming spatialization of these data will enable scientists to conduct epidemiological studies [38]. Finally, spatial indicators that include the amount of pesticides used in the vicinity or the residential distance from the fields were identified as determinants of overall exposure to agricultural pesticides through observational measurement studies. Although these data are limited and difficult to interpret, these results suggest that these two parameters are interesting surrogates for non-dietary pesticide exposure, despite their intrinsic limitations.

Developing the use of dispersion models could bring improvements to the assessment of residential exposure. GIS has difficulty assessing fine spatial variations in aerial concentrations of pesticides in a complex topographical zone. In the selected studies, GIS were generally used to define buffers around the residential location, assuming an isotropic distribution of the pesticide emissions into the atmosphere, without integrating meteorological data. One study we selected in our scoping review showed that this can result in an underestimation of populations living downwind and overestimations of residents living upwind of the treated areas [35]. Unlike GIS, dispersion models are deterministic models, based on atmospheric diffusion and reaction equations that can take into account the anisotropic distribution of the aerial dispersion of pesticides. Using modelling, it could also be possible to isolate the contribution of local emissions sources from the background emitters. Another advantage of the modelling approach is the possibility of predicting the effect of management practices on exposure or identifying the mean factors driving the exposure (by e.g. sensitivity analysis). However, some of these models demand a lot of input data (meteorological and topographical data, amounts of pesticides applied, etc.) and their computing power can limit their use in epidemiological studies involving many subjects or assessing long-term exposure. They also need to be validated with monitoring data. However, when combined with other methods such as environmental monitoring, they could exploit their advantage, as has been done in the field of assessment exposures to outdoor ambient air pollutants. Indeed, dispersion/transport models have been widely used to monitor ambient air pollution and in most epidemiological studies to study its adverse health effects [39].

Besides the methodological bias previously described, linked to the use of spatial surrogates for characterizing pesticide exposure, another source of exposure misclassification in epidemiological studies could relate to the absence of consideration of the subjects' temporal mobility. A few epidemiological studies retraced historical exposures by considering long-term residential mobility, but no publications took short-term mobility into account. However, only considering the home location and not the workplace or the school could contribute to an underestimation of global exposure. Previous studies on ambient air pollution have shown that ignoring non-domestic environments could introduce errors into the individual exposure assessment [40–42].

## Health effects linked to pesticide exposure

Finally, numerous studies found a significant association between residential exposure to pesticides and an increased risk of adverse health effects. The spatial resolution of these studies

was very variable. Some concluded that there was a positive association between pesticide exposure and health effects for short distances between the residence and the fields (< 100 m), whereas others observed an association for much longer distances (< 1000m). Scoping reviews are not able to assess the quality of the studies selected and it is not possible to interpret the meaning of these results any further. However, it does not seem possible to define a safe distance of residence from the field that could ensure the protection of human health on the basis of these results, for the same reasons that it is difficult to establish a perimeter for pesticide dispersions around an agricultural source. Indeed, according to the health effects studied, the chemicals involved are very different, as are their dispersion profiles in the environment. In the same way, Gunier et al. (2017) had previously concluded that the determination of the relevant safety distance or buffer around schools in agricultural areas should be based on the published literature of pesticide exposure and adverse health effects but currently, there is insufficient information to do so [43].

Observational measurement studies and epidemiological studies are interesting for improving knowledge about residential exposure to agricultural pesticides and its health effects. Nevertheless, the results obtained give only a partial snapshot of a situation at a particular time. The characteristics of the populations living near agricultural areas can vary greatly depending on the period and the geographical area under consideration. Agricultural practices are constantly evolving and active substances are regularly replaced in the market. Pesticides' exposure profiles can also vary over time. Risk assessment studies have the advantage of being able to include these changes and constitute an interesting management tool for regulatory and non-regulatory purposes. This domain is in constant evolution, as demonstrated by the latest publications we found.

However, a risk assessment is based on numerous assumptions that have to be discussed in each analysis. As discussed previously, the temporal and spatial evolution of pesticide concentrations in environmental compartments and the residents' exposure pathways are not yet completely characterized. This is why EFSA has recommended further production of data to produce more realistic exposure assessments in a regulatory context [4]. This also argues for further small-scale measurement studies in residential areas.

## Strengths and limitations of this scoping review

The methods implemented for exposure assessment as mapped out in this scoping review are in agreement with those determined in previous reviews [6,7]. Besides the simple description of these methods, our study offers a broader overview of the work done in assessing residential exposure to agricultural pesticides in recent years and provides new insights. The quality of our scope study lies in the fact that we used the frameworks produced by experts in the field [8–10]. A selection of relevant studies extracted from three different databases was made by at least two members of the team. Data were presented according to guidelines established by PRISMA and JBI in order to guarantee clarity in the method and transparency of the results [11,12]. We used a large algorithm to identify the relevant studies that enable us to gather a large amount of publications. However, several limitations should be noticed. Environmental monitoring or risk assessment studies conducted by scientific institutions as part of environmental health policy are usually not published in the consulted databases, and could probably not be identified despite consulting the grey literature. This is also the case for pesticide emission/dispersion models, since they are not generally used for assessing the exposure of people living near agricultural crops. However, they would provide elements of knowledge that could be used in the assessment of human exposure to pesticides. We chose to exclude studies that did not characterize the spatial organization of agricultural activities around the residence.

However, these studies could have provided information about non-spatial determinants of global exposure to residential pesticides. Finally, an inherent limitation of scoping reviews is that they do not assess the quality of the selected studies. It is not possible to interpret the level of exposure measured in the different matrices, nor the results obtained on the risk to human health of residential exposure to pesticides. Therefore, our scoping review enabled us to map the existing literature and identify its gaps in order to propose improvements in carrying out exposure assessment.

## Conclusion

The assessment of residential exposure to agricultural pesticides has become a growing area of research in the past few years. Two kinds of method have been used to assess environmental exposure, according to the study objectives: 1) direct measurements in biological and/or environmental matrices (mainly in house dust, air, and urines), predominantly used by measurement studies to identify possible determinants of exposure, and 2) indirect methods based on the establishment of spatialized exposure indicators using GIS or surveys and interviews with subjects, mostly used in epidemiological studies. Exposure in risk assessment studies was frequently obtained by modelling. It seems that pesticide exposure is largely correlated with the spatial organization of the agricultural activities in a territory. The determinants and the routes of exposure remain to be explored. Improvement of our knowledge of pesticide exposure could be expected from small-scale studies combining different exposure assessment methods, such as modelling and monitoring. Better knowledge of residential exposure would improve the conduct of epidemiological studies and risk assessments, and prevent future exposures.

## Supporting information

**S1 Appendix. References of articles selected in the scoping review.**
(DOCX)

**S2 Appendix. Table of health outcomes investigated in the epidemiological studies in the articles selected for scoping review (N = 98).**
(DOCX)

**S3 Appendix. Reporting Items for systematic reviews and meta-analyses extension for scoping reviews (PRISMA-ScR) checklist.**
(DOCX)

## Author Contributions

**Conceptualization:** Raphaëlle Teysseire, Fleur Delva.

**Data curation:** Raphaëlle Teysseire, Guyguy Manangama, Fleur Delva.

**Methodology:** Raphaëlle Teysseire, Fleur Delva.

**Supervision:** Fleur Delva.

**Validation:** Raphaëlle Teysseire, Guyguy Manangama, Carole Bedos, Fleur Delva.

**Writing – original draft:** Raphaëlle Teysseire.

**Writing – review & editing:** Guyguy Manangama, Isabelle Baldi, Camille Carles, Patrick Brochard, Carole Bedos, Fleur Delva.

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

3a1527f269290a980bf809f1f5b84f5a

