## [Decision Letter · Decision Letter 0]

5 Mar 2020

PONE-D-20-00094

Assessment of residential exposures to agricultural pesticides: a scoping review

PLOS ONE

Dear Mrs Teysseire,

Thank you for submitting your manuscript to PLOS ONE. After careful consideration, we feel that it has merit but does not fully meet PLOS ONE’s publication criteria as it currently stands. Therefore, we invite you to submit a revised version of the manuscript that addresses the points raised during the review process.

We would appreciate receiving your revised manuscript by Apr 19 2020 11:59PM. After reviewing the article myself I feel some minor points may be addressed in the paper before final publication. To increase the speed of the review process, I attached a word document that the authors may see as the secund review of this paper. It may also be addressed in the rebuttal letter.

To enhance the reproducibility of your results, we recommend that if applicable you deposit your laboratory protocols in protocols.io, where a protocol can be assigned its own identifier (DOI) such that it can be cited independently in the future. For instructions see: http://journals.plos.org/plosone/s/submission-guidelines#loc-laboratory-protocols

We look forward to receiving your revised manuscript.

Kind regards,

Pieter Spanoghe

Academic Editor

PLOS ONE

Journal Requirements:

Please ensure that your manuscript meets PLOS ONE's style requirements, including those for file naming. The PLOS ONE style templates can be found at http://www.plosone.org/attachments/PLOSOne_formatting_sample_main_body.pdf and http://www.plosone.org/attachments/PLOSOne_formatting_sample_title_authors_affiliations.pdf

Reviewers' comments:

Reviewer's Responses to Questions

**Comments to the Author**

1. Is the manuscript technically sound, and do the data support the conclusions?

Reviewer #1: Yes

2. Has the statistical analysis been performed appropriately and rigorously? 

Reviewer #1: N/A

3. Have the authors made all data underlying the findings in their manuscript fully available?

Reviewer #1: Yes

4. Is the manuscript presented in an intelligible fashion and written in standard English?

Reviewer #1: Yes

5. Review Comments to the Author

Reviewer #1: I think the paper gives a good overview.

Some details:

- line 46: is there a more international publication (e.g. sth from OECD of WHO) instead of (2)? Is a citation actually needed?

- line 50: ref (3) does not seem to be the most dedicated paper pathways.

- lines 53-56: How do you conclude from only paper (4) what is said about farmworkers vs residents?

- line 58-59: who concluded there is no consensus?

- line 77: OPEX guidance does not say "only if the risk to humans can be considered as negligible" in these words. Also, would ref (1) not be a more precise source for this?

- Lines 77-79: Are two messages mixed here? "including monitoring of the environment": is monitoring a preventive measure? Or, is understanding of residents' pathways and levels of exposure essential for monitoring of the environment (for the sake of the environment)??

- line 335-336: title does not seem to cover the sections' messages. Reword?

- lines 418 - 449: is this section needed? Health effects as such are not the focus. The part devoted to Gunier et al (2017) is not very clear to me. It has to do with identifiying risk mitigation measures, but I am not sure why this study is highlighted primarily on its conclusions. I think there is much more information on risk reduction measures (distance, barriers) out there.

In general: how many observational studies recorded actual applications in relation to the observations?

Under Strenght and weaknesses: Check the grey literature on the Dutch resident study: https://www.bestrijdingsmiddelen-omwonenden.nl/documenten/onderzoeksrapport-obo-1.

6. PLOS authors have the option to publish the peer review history of their article (what does this mean?). If published, this will include your full peer review and any attached files.

Reviewer #1: Yes: Mark HMM Montforts

---

## [Author Response · Author response to Decision Letter 0]

30 Mar 2020

We thank the academic editor and the reviewer for their valuable comments and suggestions. We have made changes to the article in an effort to make it more intelligible. The modifications are described in a word document, and we give a point-by-point response to the reviewers’ concerns.

---

## [Editor Report · Decision Letter 1]

13 Apr 2020

Assessment of residential exposures to agricultural pesticides: a scoping review

PONE-D-20-00094R1

Dear Dr. Teysseire,

We are pleased to inform you that your manuscript has been judged scientifically suitable for publication and will be formally accepted for publication once it complies with all outstanding technical requirements.

With kind regards,

Pieter Spanoghe

Academic Editor

PLOS ONE
---

## [Editor Report · Acceptance letter]

15 Apr 2020

PONE-D-20-00094R1 

Assessment of residential exposures to agricultural pesticides: a scoping review 

Dear Dr. Teysseire:

I am pleased to inform you that your manuscript has been deemed suitable for publication in PLOS ONE. Congratulations! Your manuscript is now with our production department. 

With kind regards,

on behalf of

Prof. Dr. Pieter Spanoghe 

Academic Editor

PLOS ONE